

# Characterization of refractory aerosol particles collected in the tropical UTLS within the Asian Tropopause Aerosol Layer (ATAL)

Martin Ebert[1]*, Ralf Weigel[2], Stephan Weinbruch[1], Lisa Schneider[1], Konrad Kandler[1], Stefan Lauterbach[1], Franziska Köllner[2,4], Felix Plöger[3], Gebhard Günther[3], Bärbel Vogel[3], and Stephan Borrmann[2,4]

[1] Institut für Angewandte Geowissenschaften, Technische Universität Darmstadt, Germany
[2] Institut für Physik der Atmosphäre, Johannes Gutenberg-Universität, Mainz, Germany
[3] Institut für Energie und Klimaforschung (IEK-7), Forschungszentrum Jülich, Germany
[4] Partikelchemie, Max-Planck-Institut für Chemie, Mainz, Germany

*Correspondence to*: Martin Ebert (mebert@geo.tu-darmstadt.de)

**Abstract.** Aerosol particles with diameters larger than 40 nm were collected during the flight campaign StratoClim2017 within the Asian Tropopause Aerosol Layer (ATAL) of the 2017 Monsoon Anticyclone above the Indian subcontinent. A multi-impactor system was installed on board of the aircraft M-55 Geophysica, which was operated from Kathmandu, Nepal. The size and chemical composition of more than 5000 refractory particles/inclusions of 17 selected particle samples from 7 different flights were analyzed by use of scanning electron microscopy (SEM) and transmission electron microscopy (TEM) combined with energy dispersive X-ray microanalysis (EDX). Based on chemical composition and morphology, the refractory particles were assigned to the particle groups: extraterrestrial, silicates, Fe-rich, Al-rich, Hg-rich, other metals, C-rich, soot, Cl-rich, and Ca-rich.

Most abundant particle groups within the refractory particles are silicates and C-rich (nonvolatile organics). In samples taken above the tropopause extraterrestrial particles are becoming increasingly important with rising altitude. The most frequent particle sources for the small (maximum in size distribution $D_{P-max}$ = 120 nm) refractory particles carried into the ATAL are combustion processes at ground (burning of fossil fuels / biomass burning) and the agitation of soil material. The refractory particles in the ATAL represent only a very small fraction (< 2 % by number for particles > 40 nm) of the total aerosol particles which are dominated by species like ammonium, sulfate, nitrate, and volatile organics. During one flight additionally a large number of very small ($D_{P-max}$ = 25 nm) cinnabar particles (HgS) were detected. These particles are most likely generated directly on ground by coal combustion in Northeastern India or Southern China.

These findings show that coal burning is an important source for the entry of refractory particles and in particular mercury into the ATAL respectively in the upper troposphere/ lower stratosphere (UTLS) region.

## 1 Introduction

Each year during the summer monsoon from June to September, the Asian tropopause aerosol layer (ATAL) develops inside the Asian monsoon anticyclone (AMA) (Vernier et al., 2018; Zhang et al., 2019). The ATAL forms in the upper troposphere/ lower stratosphere (UTLS) region at altitudes of about 14 to 18 km corresponding to potential temperature levels of 360K to

420K (Hanumanthu et al., 2020). The lateral dimensions of the AMA can extend from the eastern Mediterranean up to East Asia (Vernier et al., 2011).

During the summer monsoon large-scale convection in the Himalayan region provides a strong upward transport of gases and particles from the ground to the ATAL. Because of "eddy shedding" (Dethof et al., 1999; Popovic et al., 2001; Pan et al., 2016), a smaller fraction of the particles is even transported into the lower stratosphere and can there be subjected to long-

range transport (Vogel et al., 2016 & 2019; Fujiwara et al., 2021).

Model analysis by Fairlie et al. (2020) showed the dominance of regional anthropogenic emissions of particle precursors like sulfate, nitrate, ammonia and organic aerosol particles from China and the Indian subcontinent in affecting observed aerosol concentrations in the ATAL. The first in-situ mass spectrometric analysis of aerosol particles within the ATAL (Appel et al., 2022) determined that the particles in the ATAL consist mainly of ammonium nitrate and organics. It was further found that

up to 70% of these are formed from the conversion of inorganic and organic gas-phase precursors rather than from a direct uplift of primary particles from the boundary layer. Höpfner et al. (2022) gained important insights into the formation of ammonium nitrate from agricultural emissions of ammonia uplifted within the Asian monsoon in the UTLS region.

But still, all details of processes involved in nucleation of inorganic compounds (including sulfuric acid, ammonium salts and nitric acid) as well as secondary organic aerosol (SOA) formation in the AMA are not completely understood. It is also not

clear whether aerosol particles from the AMA (either transported from below or newly formed within the AMA) are a relevant source of the stratospheric Junge layer (Appel et al., 2022).

An increased particle concentration has a variety of severe atmospheric implications, including the high relevance for the climate system. First, aerosol particles within the ATAL can directly influence /affect the radiative budget at the top of the

atmosphere (Vernier et al., 2015). Second, aerosol particles inside the AMA are involved in ice-cloud formation below the tropopause and in the tropical transition layer (Wagner et al., 2020, Ueyama et al., 2018). They can act as heterogeneous ice nuclei influencing cirrus cloud properties (see e.g., Liu et al., 2009; Fadnavis et al., 2013). Third, those aerosol particles can It was shown in modeling studies that some of UTLS particles in the AMA should include refractory components (Fadnavis et al., 2013; Lau et al., 2018; Ma et al., 2019), eventhough the main components are sulfuric acid/sulfates, nitric acid/nitrates,

water, ammonium and organic compounds (Appel et al., 2022; Höpfner et al., 2019).

The physicochemical properties of refractory particles (concentration, chemical composition, size, mixing state) are important for model calculations of all the processes mentioned above. In UTLS aerosol particle studies conducted outside the AMA, most refractory particles are assumed to originate from meteoric ablation, space debris, or rocket exhaust (e.g., Borrmann et al., 2010; Murphy et al., 2014; Ebert et al., 2016; Schneider et al., 2021). With the exception of volcanic eruptions and wildfires,

the transport of refractory particles from the Earth`s surface is considered to be of minor importance (Kremser et al., 2016).

For the ATAL, vertical transport of gases and particles from the boundary layer plays a much larger role. Based on the investigation of the transport pathways and the dynamics in the tropics, several studies conclude that the most important refractory particles in the ATAL are mineral dust, black carbon, metal(oxides), and to a smaller extent meteoric material (e.g., Lelieveld et al., 2018; Lau et al., 2018; Ma et al., 2019, Bossolasco et al., 2021). A few studies even state that the main

constituents within the ATAL are mineral dust and black carbon aerosol (Bossolasco et al., 2021; Ma et al., 2019). On the Indian subcontinent, numerous anthropogenic sources exist for small refractory particles which have a low enough inertia to enable transport to such high altitudes (e.g., Lawrence and Lelieveld, 2010).

Some of the particles originating on ground will experience scavenging and chemical processing during transport through

convective mixed-phase clouds. As a consequence, the chemical composition of aerosol particles in the ATAL may differ from the ground conditions (Froyd et al., 2009; Jost et al., 2017). However, experimental data on physicochemical properties of

refractory particles within the ATAL and their sources is sparse (Vernier et al., 2022). This contribution aims on improving the experimental data base in order to gain a better understanding of particle transport into the ATAL.

Please be aware that different definitions of the term "refractory" and "non-volatile" are used in the literature. For example, during StratoClim 2017 Mahnke et al. (2021) used a multi-channel condensation particle counter (COPAS) and an optical particle spectrometer (Ultra High Sensitivity Aerosol Spectrometer UHSAS-A) for detecting total aerosol densities of submicrometer sized particles in the ATAL. In these measurements particles are classified as "non-volatile" which have passed through a at 270°C heated tube section within COPAS of about one meter length.

In this way, in these measurements some of the secondary sulfate, nitrate and organic particles could be classified as non-volatile (i.e. thermostable at up to 270°C).

In this work, the term refractory refers to all particles stable at electron bombardment under high vacuum conditions in the scanning electron microscope.

## 2 Experimental

### 2.1 Flght campaign StratoClim 2017

Within the EU Framework Programme 7 project StratoClim (Stratospheric and upper tropospheric processes for better climate predictions) a stratospheric aircraft campaign (StratoClim 2017) was conducted in July-August 2017 in Kathmandu, Nepal. One main goal was the characterization of the Asian tropopause aerosol layer (ATAL) within the 2017 monsoon anticyclone. In total, 8 flights were performed by the Russian high-altitude aircraft M55-Geophysica. The aircraft was equipped with a variety of in-situ and remote sensing equipment for the measurement of particle and gas composition. An overview can be found in Stroh et al. (2023). The flights took place every second day during the period 27 July – 10 August. Sampling details can be found in Table 1.

The flights were conducted from Kathmandu (Nepal) Tribhuvan International Airport (TIA) with a total flight time of about 31 h. Three flights (#2, #4, #5) took place exclusively above Nepal. These flights were carried out along an axis parallel to the Himalaya over almost the entire east–west extension of this country. Three further flights (#3, #7, #8) were performed over northeastern India. These flight patterns allowed the study of the horizontal structure of the AMA over large parts of its north–

south extension, although the flight tracks did not reach out of the anticyclone core (von Hobe et al., 2021). For more details, see Khaykin et al. (2022).

According to Bucci et al. (2020) and Brunamonti et al. (2018), the first half of the StratoClim 2017 campaign (# 1 - 4) was

less affected by regional convective activity than the second half (# 5 - 8). The minimum and maximum flight height during the particle collection periods of the particle samples studied in detail are given in Table 1. The absolute potential temperature (Θ) throughout sampling based on ambient condition data (air temperature and static pressure from aircraft UCSE) during the sampling period are given as boxplots in Figure 1 (upper part). In both parts of Figure 1, the boxes represent the lower and upper quartiles. A horizontal black line within the box marks the median, a horizontal red line the mean. Whiskers below and

above the box indicate the 10th and 90th percentiles. Crosshair symbols represent the 5th and 95th percentiles.

The boxplot illustration in Figure 1 (lower) illustrates the potential temperature (Θ) difference to the 1 Hz calculated Θ-level of the cold point tropopause (CPT) during the sampling period. Positive (negative) ΔΘ indicate a sampling above (underneath) the CPT. The CPT-potential temperature is extracted from ERA interim data (Weigel et al., 2021a). The potential temperature (Θ) was calculated based on UCSE data of ambient temperature and pressure as defined by the World Meteorological

Organization (WMO, 1966). For the vertical temperature and pressure distribution during the impactor collection phases in StratoClim 2017, the WMO-compliant Θ values deviate by no more than ~ 1 K from the results according to the refined Θ calculation (Baumgartner et al., 2020).

## 2.2 Sampling technique

Particle samples were taken by the inlet line of COPAS (Condensation Particle counting System; Curtius et al., 2005; Weigel et al., 2009; Borrmann et al., 2010) with a Y-shape manifold. According to Weigel et al. (2009) the inlet efficiency is comparable with the inlet system characterized by Hermann et al. (2001). For submicron particles, the transmission efficiency of the COPAS inlet is ≥ 90 %. The inlet performance rapidly deteriorates for increasing particle diameters and is between 30 – 40 % for particles with 4 µm diameters, and ≤ 5 % for particles larger than 6 µm.

During the campaign a multi cascade impaction system Multi-MINI (Ebert et al., 2016) was operated downstream of the

COPAS inlet. Twelve dual stage impactors are integrated into a single housing and particle sampling of the single impactors

is controlled by a set of valves. A 12-fold symmetrical manifold directs the aerosol to the separate units.

The orifices of the individual dual stage impactors are 0.75 mm and 0.25 mm in diameter. Air velocity in the second nozzle is

at speed of sound and, thus, controlling the impactor flow, which was calculated to be around 7.7 $cm^3$/s. During UTLS

sampling, temperature in the COPAS system varied between 272 K and 290 K, pressure between 50 and 67 hPa.

At these conditions in the UTLS fifty percent efficiency cut-offs (calculated according to Raabe et al., 1988) are ~ 400 nm

aerodynamic diameter for the first impactor stage, and ~ 40 nm for the second. Please note, that under UTLS conditions the

strictness of the impactor size discrimination is inferior compared to tropospheric conditions.

In this study, particle samples of the first impactor stage are referred to as coarse fraction (> 400 nm), those of the second stage

as fine fraction (~40 - 400 nm).

A purge flow system is added to the Multi-MINI which floods the tubing and the interior of the manifold prior to each sampling

with ambient air to avoid any carryover of particles from previous measurements. The purge flow extends to the front of the

first impaction nozzle, so the potential volume affected by carryover is minimal. The purge time (7 minutes) was chosen so

that the tube and manifold volume could be filled at least ten times with the current aerosol. Such a purge flow system has

proven to be crucial in UTLS aerosol particle sampling in order to minimize sampling artefacts above all a carryover from the

boundary layer (Ebert et al., 2016).

During StratoClim 2017, the separate impactors were operated in the UTLS for 13 - 18 minutes each (Table 1). This sampling

time was a compromise between receiving a sufficient number of refractory particles and simultaneously avoiding an

overloading of the sampling substrate by the dominating semi-volatile sulphate, organic and or nitrate particles, which would

hinder accurate electron microscopic analysis of the much smaller number of refractory particles. Further on, the chosen

sampling time allowed us to collect up to 6 UTLS particle samples during each flight.

Particles were collected on Ni TEM grids (S162-N9, Plano GmbH, Wetzlar, Germany). A total of 42 dual stage impactor

samples were collected during 7 mission flights. The first flight of the campaign (27[th] of July) was exclusively used for testing

the sampling setup including blind sampling for detection and elimination of possible particulate artefacts.

The main goal of the present study is the physicochemical characterization and source apportionment of refractory particles

within the ATAL. These non-volatile particles are expected to be very small (< 500 nm) and to occur in very small numbers.

Furthermore, they will be often embedded in or agglomerated with the dominating secondary sulphate, nitrate and/or organic

particles. Please note that the analyzed refractory inclusions can be much smaller than the lower cut-off diameter of the

sampling device.

It should be emphasized here again that the term "refractory" is used in the present paper for all particles which are stable

during electron bombardment under the high vacuum conditions of the electron microscope in contrast to the sulphate-,

organic- and nitrate-containing particles, which evaporate quickly.

**2.3 Sampling and Analysis Strategy**

A major challenge of particle sampling in the UTLS region is avoiding sampling artefacts which can be caused by abrasion

within the aircraft sampling line (manifold/inlet/collector) or by carryover of particles from the boundary layer.

As the particle concentrations in the UTLS are very low and refractory particles/inclusions represent the smallest share, even

a small contribution of refractory artifacts will severely distort the results. In order to minimize the risk of artefacts, impactor

sampling has the advantage that all collected particles are deposited within a very small area (impaction spot) on the sampling

substrate. Since TEM substrates are almost particle-free before sampling, and in impactor collection the impaction spot on the

sampling substrate is very small ($<<$ 1mm$^2$), the number of artefact particles is negligible, if the number of collected particles

within the impaction spot is high and no artefact particles originate from the sampling line itself. In several procedural blank

tests (e.g., complete flight 1) no particulate artifacts were detected in samples after the rinsing unit was used.

Nevertheless, for any individual ambient collection there is still the risk of artefact introduction into the samples due to individual events during installation, in-flight, or during removal and transport of the particulate samples.

To minimize the risk of interpreting artefact particles as real refractory components, only samples that met two additional criteria upon first inspection were selected for analysis.

First, all carriers which show any kind of particulate contamination upon first inspection were excluded from further analysis. Contamination is confirmed when particles are found on the sampling substrate that have a steel-like composition or are too large to be sampled within the UTLS. Only few samples had to be excluded based on this criterion. Second, samples were excluded when too few particles were found on the substrate as in this case it cannot be guaranteed that the number of potential refractory artefact particles is negligible. Based on this criterion, a large number of the received particle samples from flights

#2, #3, #4, #5, and #6 had to be excluded. In addition, samples with less than 25 refractory particles found during the analysis step were also excluded.

In this way, only 17 out of 84 received particle samples were investigated in detail. Six of these samples were fine stages and eleven coarse stages. In 5 cases it was possible to analyze fine and coarse stage pairs of the same sample (#5.2, #7.1, #7.4,

#8.1, and #8.2).

## 2.4 Characterization of refractory particles/inclusions by electron microscopy

Individual particle analysis was performed in a FEI (Eindhoven, the Netherlands) Quanta 200 FEG Environmental Scanning Electron Microscope (ESEM) equipped with an energy-dispersive X-ray detector (EDX, EDAX, Tilburg, Netherland). As the instrument was operated under high vacuum conditions only, we will refer to the method as scanning electron microscopy (SEM) throughout the paper.

A detailed analysis of the dominating sulfates and nitrate particles was not intended. Instead, we focus on the detection of

refractory particles/inclusions. In a first step many refractory particles were detected using backscattered electron (BSE)

imaging by their higher average atomic number (leading to higher brightness) compared to the dominating sulfate, nitrate and organic species.

This procedure was successful in detecting externally mixed high-Z refractory particles, but low-Z refractory particles (dominantly carbonaceous particles) and completely embedded refractory inclusions will stay undetected.

Therefore, several thousands to tens of thousands volatile particles in each sample were evaporated in the instrument by electron bombardment. In this way, refractory inclusions were detected in $1 - 2$ % of the volatile particles.

This time-intensive analytical step was also necessary to provide a statistically significant number of refractory particles for individual particle samples. Using this approach, it was possible to analyze a significant number of refractory particles/inclusions ($28 - 741$ particles per sample) in seventeen flight samples (Table 2).

Additional measurements were performed in a Jeol 2100 Transmission electron microscope (TEM) which was equipped with an Oxford INCA EDX system to determine the mineralogical phase of nm-sized Hg-rich particles.

**3 Results**

**3.1 Refractory particles/inclusions**

Even when refractory particles play an important role in many atmospheric processes in the UTLS (see introduction) they only account for a small fraction of the total aerosol population in the ATAL. In this SEM/EDX study heterogeneous inclusions of

refractory components were observed in around 2% of the analysed ATAL particles.

During StratoClim 2017, simultaenous aerosol mass spectrometric measurements with the ERICA-LAMS (Laser Ablation Mass Spectrometer) instrument were conducted. ERICA-LAMS is able to measure refractory aerosol components by laser ablation and ionization technique followed by time-of-flight aerosol mass spectrometry (Hünig et al., 2022; Dragoneas et al., 2022). In these measurements it was determined that between 20 and 50 % of all measured particles (by number) include

refractory material, depending on altitude (Appel et al., 2022). The values between the two techniques differ, obviously, for different reasons. The refractory number abundance of 2% determined by SEM/EDX measurements refers to refractory

heterogeneous inclusions within ATAL particles only (see chapter 2.4) while the MS derived value of 20 – 50 % reflects the proportion of particles that provide any refractory signal including dissolved refractory elements, which may play also an important role (e.g. for extraterrestrial material, see Schneider et al., 2021 and discussion in chapter 4.3).

Contrary to earlier modeling studies (Fadnavis et al., 2013; Lau et al., 2018, Ma et al., 2019, Bossolasco et al., 2021), we found that refractory particles (including desert dust) make up the minority aerosol components in the ATAL.

In all samples, the secondary components (sulfate, nitrate, and organic aerosol particles) are the dominant particle types. These mainly volatile species, however, are not in the scope of this work and are not regarded further. Detailed data on the concentration and distribution of volatile main species of UTLS particles during StratoClim 2017 can be found in Höpfner et

al. (2019), Yu et al. (2022) and Appel et al. (2022).

In total, 5033 refractory particles/inclusions were detected within the 17 selected UTLS particle samples. Based on EDX spectra and morphological criteria these refractive particles/inclusions were classified into 10 particle groups: silicate, extraterrestrial, Ca-rich, Cl-rich, Fe-rich, Al-rich, other metals, soot, C-rich, and Hg-rich. All refractory particles, which fit in

none of these groups were summarized in an eleventh "other" group. The absolute number of detected particles for each group and flight sample is given in Table 2, their relative abundances in Figure 2.

The smoothed relative size distributions of 5 refractive particle groups are plotted in Figure 3. Because of the limited number of particles this distribution can only be shown for the most abundant refractory particle groups.

Approximately 30% (1499 out of 5033) of refractory particles were Hg-rich particles with very small diameters (maximum of

size distribution $D_{P-max}$ = 25 nm). These particles occur almost exclusively in all samples of flight 8. Based on the absolute particle numbers, it is assumed that the Hg-rich particles are an additional load. Since relative abundances are compositional data (i.e., they have a constant sum) the comparison of samples from different days can be misleading. Therefore, the Hg-rich particles were excluded in the further comparisons of the relative particle abundance in this chapter and are discussed separately (chapter 4.4).

For particle classification all element peaks except Ni and S derived from EDX spectra were used. Ni is often present in the EDX spectra because of the used Ni grid sample substrate. As most of the refractory UTLS particles are either embedded or agglomerated to sulphate/nitrate/organic particles, sulphur is detected in many particle spectra without giving an indication

whether the element peak is originating from the refractory particle itself or the surrounding sulphate matrix. It must be noted that because of the small size of most detected refractory particles ($D_P$ < 100 nm for 50 % of all detected refractory particles) only the major elements of such small particles can be detected by EDX, minor elements are often not clearly distinguishable from the spectrum background.

*Extraterrestrial material*

Mg-rich silicates as well as Mg- and Fe-rich particles were classified as extraterrestrial (chondritic composition as proposed by Rietmeijer, 1998). Following the Rietmeijer classification, the extraterrestrial group has an average relative abundance of 11 % within the refractory particles (2 – 40 % in the individual samples). Average particle diameter ($\overline{D}_P$) was 290 nm.

*Silicates*

All particles with Si and O as major elements, but without Mg were classified as silicates. As minor elements often Na, Al, K, Ca or Fe were found. In total, 41 % of all detected refractory particles were classified as silicates. They were found in all 17 samples with an abundance between 21 and 58 %, which makes these particles to the most abundant refractory particle group. $D_{P-max}$ for the silicates was found to be 120 nm ($\overline{D}_P$ = 170 nm).

*Ca-rich*

All particles with Ca and O as major elements were classified as Ca-rich. Additionally, carbon was found in most of these particles as main element.

The abundance of the Ca-rich particle group is mostly low. In only 5 of the 17 samples this group contributes to more than 2 % of the detected refractory particles (maximum 7 %). $\overline{D}_P$ of the Ca-rich particles is 210 nm.

*Cl-rich*

Particles with Cl as main peak were classified as Cl-rich. Besides Cl only carbon and oxygen were detected as main peaks. Single Cl-rich particles were detected in 15 of the 17 samples but always with very low abundances (0 - 4 %). These particles have with 510 nm the largest $\overline{D}_P$ of all refractory particle groups.

*Fe-rich*

Particles with Fe and O as main peak (without detectable Mg) were classified as Fe-rich. Most of these particles show no other metal peaks, only in single particles very small peaks of Al, Cr or Mn were detected. Fe-rich particles were observed in all 17 particle samples with an average abundance of 9 % (2 – 30 %). $\overline{D}_P$ of Fe-rich particles was 250 nm.

*Al-rich*

Aluminum and oxygen rich particles were classified as Al-rich.  These particles play only a minor role in UTLS and only some individual particles were detected (average 1%, range 0 - 4 %). $D_{P\text{-max}}$ was found at 125 nm ($\overline{D}_P = 250$ nm).

*Other metals*

Besides Al- and Fe-rich particles, about 300 refractory particles of different metals (or metal oxides) were detected and summarized in the "other metals" group. Most of these particles were Mn-rich (78), Cr-rich (71), Zn-rich (63), Sn-rich (26),

Pb-rich (20), W-rich (17) or Ti-rich (14). Additionally, a few individual particles of Ba, La, Mg, Sb, Cu and Ce were found. $\overline{D}_P$ of the other metals group was 290 nm.

*C-rich and Soot*

All refractory particles, which show only carbon and oxygen peaks were classified as soot or C-rich. Soot can be recognized in SEM by its typical morphology of agglomerates of spherical primary particles. As the lateral resolution of the SEM is

limited, this morphological criterion cannot be applied to small particles with diameters < 50 nm. Thus, the abundance of the soot group represents a minimum share. Soot particles were detected in 13 of the 17 samples.  $D_{P\text{-max}}$ and $\overline{D}_P$ of soot particles were determined at 250 nm. All refractory carbon rich particles, which could not clearly be identified as soot were summarized in C-rich. C-rich particles (or non-volatile organic compounds NVOC) were detected in all 17 samples with an average abundance of 17 % (3 - 41 %). $\overline{D}_P$ of the C-rich group was 180 nm. Some very small soot particles will not be recognized

accurately in SEM analysis and will be classified as C-rich.

## 4. Discussion

### 4.1 Sources of the refractory particles in AMA

*Extraterrestrial*

For the classification of stratospheric particles often a broader classification is applied as it was used here. For example, in the NASA cosmic dust catalogue (Warren et al., 2011) not only chondritic compositions are classified as "cosmic", but also compositions strongly modified by ablative heating or melting during passage through the atmosphere. Following the NASA definition (which is not complete applicable for this work as it is defined for optical microscopic data of large super-µm particles) all particles from our extraterrestrial-, silicate-, and Fe-rich groups would be classified as "cosmic". The NASA definition is appropriate for stratospheric particles, collected well above the tropopause, where little terrestrial admixture is present. For particles collected below the tropopause and specially inside the ATAL, where increasing entry of terrestrial silicate material occurs, an unambiguous attribution of Mg, Si and/or Fe-containing particles to a terrestrial or extraterrestrial source is difficult. Thus, the classification of extraterrestrial particles in the upper troposphere used here is associated with a substantial uncertainty.

Furthermore, it has to be considered that after ablation processes a part of the incoming extraterrestrial material (iron) may be "dissolved" in sulfuric acid droplets (discussion in chapter 4.3). If the total extraterrestrial input should be estimated, both, the dissolved fraction and the refractory particles/inclusions have to be considered.

*Silicates*

The particles of the silicate group (Si and O rich / Mg-free) show the typical element signatures of terrestrial silicates with minor elements, as for example Na, Al, K, Ca or Fe.

Soil, coal burning and modified extraterrestrial material are the three most important sources for the silicates. Only the smallest soil particles managed transport into the UTLS, while the larger ones sedimented before due to their inertia.

*Ca-rich*

As most of the Ca-rich particles also contain C and S as major elements (sometimes Mg and K as minor elements) they are interpreted as calcium carbonates (calcite/dolomite) or calcium sulphates (gypsum/anhydrite). The main sources of these Ca-rich particles are soil and industrial combustion processes such as coal and fossil fuel burning.

*Cl-rich*

Particles containing only Cl, C and O as main elements are interpreted as an organochlorine compound from industrial or secondary processes. An internal mixture of sea-salt and organic particles is also conceivable, but unlikely as Na and Mg was not detected in any of these particles.

*Fe-rich*

While the highest observed abundance of the Fe-rich group for samples from flight 2-7 was 11 % (range 2 - 11 %), in samples

of flight 8 significantly higher abundances up to 30% (range 7 - 30 %) were encountered. Since flight 8 is characterized by a

strong updraft (chapter 4.3 / 4.4), and there is an increased input of terrestrial refractory particles in these samples, it is assumed

that a large fraction of these Fe-rich particles stems from terrestrial sources, most likely due to industrial high temperature

processes (e.g. in forges or smelters) and burning of fossil fuels. However, the terrestrial/extraterrestrial origin of Fe-rich

particles in the UTLS is a subject of current debate and an extraterrestrial origin of single Fe-rich particles cannot be excluded

(Ebert et al., 2016).

*Al-rich*

Al-rich particles are supposed to be mainly aluminum oxide and originate predominantly from solid rocket fuel exhausts

(Mackinnon et al., 1982; Cziczo et al., 2002). Al-rich particles in the stratosphere and their impact on stratospheric ozone were

studied since the early 70s (Hoshizaki, 1975; Denison et al., 1994; Jackman et al., 1998; Danilin et al., 2001).

Cofer III et al. (1991) measured a bimodal size distribution of aluminum oxide particles in the Space Shuttle plume with peaks

at <0.3 and 2 µm, while in this study no super-µm Al-particles were detected.

*Other metals*

Anthropogenic high-temperature processes and burning of fossil fuel are assumed to be the main sources of the diverse

metal/metal oxide particles (Mn, Cr, Zn, Sn, Pb, W, Ti) detected in this study.

*Soot*

During data analysis, the assumption that the abundance of the small refractory particles/inclusions would be the same in the

coarse and fine stages was disproved for the soot group. Soot is observed at higher abundances on the fine stages (3 - 22 %) in

contrast to the coarse stages (0 - 5 %). As soot agglomerates have a very small aerodynamic diameter (small $D_P$ and low density

respectively a high pore volume) and - in contrast to all other refractory particles - they were often not embedded within larger

particles, much higher abundance of these particles was found on the fine stage. Fossil fuel burning, industrial processes and

traffic are the most likely terrestrial sources. However, no strong enrichment of soot was observed for flights 7 and 8 which

are characterized by significant updraft.

*C-rich*

The sources for the small refractory carbon-rich particles respectively NVOCs are nucleation processes. Sources of such nucleation particles can be either natural or anthropogenic primary emissions at ground (e.g., fossil fuel burning) as well as

secondary atmospheric processes. During the StratoClim aircraft campaign in 2017 it was detected that organics in general along with ammonium, sulfate, and nitrate are the main constituents of the ATAL (Yu et al., 2022).

## 4.2 Absolute concentration of refractory particles in AMA


All abundances of refractory particles discussed so far are relative proportions. Estimation of absolute concentrations is associated with large uncertainties. The two main uncertainties are the poorly known collection efficiency in the airborne particle collection system under extreme and variable ambient conditions and the inhomogeneous deposition of the secondary particles (splattering) on the TEM grids. Therefore, the main discussion is focused on the relative proportions in order to avoid

misinterpretations.

Nevertheless, some quantitative statements can be made about the proportion of refractory particles in relation to the dominant sulfate/nitrate matrix. During StratoClim 2017 a total of 5033 refractory particles/inclusions within 270.000 analyzed volatile sulfate/nitrate/organic particles was found (~ 1.9 % by number). The total particulate concentration in ATAL during StratoClim 2017 was on the order of 1 - 2 µg/m$^3$ (Appel et al., 2022; Yu et al., 2022; Höpfner et al., 2019). Taken into account the small

size of most refractory particles/inclusions the total concentration of all refractory particles in the ATAL above Nepal during StratoClim 2017 will be < 10 ng/m$^3$, most probably often even ≤ 1 ng/m$^3$.

## 4.3 Variability of the relative abundance of refractory particles in AMA

For the whole StratoClim 2017 campaign the silicate group was the main refractory particle group (41 %), followed by

carbonaceous particles (17 %), and extraterrestrial particles (11 %). Fe-rich particles and the "other metals" group occur also at higher relative abundance (8 % each), while only minor portions (1 – 4 % each) were determined of the Ca-rich, chloride,

Al-rich and soot groups. All percentages given above are calculated without Hg-particles (see chapter 3.1), which are discussed separately (chapter 4.4).

Only a small fraction of refractory particles emitted at ground reaches the lower stratosphere, and only a small fraction of
extraterrestrial particles the AMA. This can be seen from the average ratio of ground-emitted / extraterrestrial refractory particles, which drops from 16.6 in all samples collected below the tropopause (2743 ground emitted particles / 165 extraterrestrial particles) to 4.2 in all samples collected above the tropopause (549 ground emitted particles / 131 extraterrestrial particles). During StratoClim 2017 terrestrial particles dominate the composition of refractory particles clearly in the ATAL and slightly above the tropopause.

In our measurements the relative number abundance of the extraterrestrial particles decreases from 19.3 % above the tropopause to 5.7 % below the tropopause (within the refractory particles). In the same campaign, a substantial decrease from 13.3 % to 0.3 % was also detected by Schneider et al. (2021) by single particle mass spectrometry (MS) with the ERICA-instrument. A variety of stratospheric MS measurements have detected signatures of elements from meteoric material within a significant fraction of the dominant sulfuric acid / sulfate particles (Murphy et al., 1998 & 2014; Cziczo et al., 2001; Froyd
et al., 2009, Schütze et al., 2017).

However, the results of Schneider et al. (2021) and our measurements cannot be compared directly due to the different size range analyzed and the different sensitivity of the two analytical techniques. Further on, not the complete extraterrestrial material detected by ERICA will be present in the form of heterogeneous refractory particles. Instead, after ablation some fraction of the extraterrestrial material may be dissolved within sulfuric acid droplets (Kremser et al., 2016). For example,
Murphy et al. (2014) assumed that Fe, Ni, and Mg within UTLS particles is dissolved, while Si and Al may be present in form of refractory solid inclusions. Fe and Mg occurring in small amounts within sulfate particles collected at 2 - 8 km height in the vicinity of a tropopause fold over the Western Pacific in 2013 were also interpreted as dissolved components originating from meteoric ablation (Adachi et al., 2022).

The presence of Fe and Mg in dissolved form would explain the fact that in our SEM study only a very small number of solid
Mg-rich Fe particles was found, and that the particles classified as extraterrestrial mainly consist of Mg-rich silicates. Furthermore, this partial dissolution of some meteoric elements will also strongly modify the composition of the resulting



silicates (in contrast to the composition of original meteoric material). Thus, it is possible that some of the particles classified as silicates in this study will have an extraterrestrial origin. At least in the samples collected above the tropopause a relevant part of the silicate particles may be residuals of ablation processes, even when it is not possible to differentiate them from

terrestrial silicates by their main elemental composition.

The relative abundance of all terrestrial refractory particle groups shows quite low variability within the 17 different flight samples regardless of the specific flight altitude. This is remarkable as some samples were collected at lowermost levels of the ATAL (e.g., sample 2.1 in 12.5 – 15 km), some samples well within the ATAL (just below the tropopause) and some samples

significantly above the tropopause (e.g., sample 3.6 in the free stratosphere at 19.8 km). This implies that a small fraction of the refractory particles emitted at ground was transported into the lower stratosphere in the Asian monsoon region in 2017. This is consistent with results of $CO_2$ measurements during StratoClim, showing that during the Asian Monsoon spatio-temporal patterns of $CO_2$ on the Indian Subcontinent driven by

regional flux variations rapidly propagate to approximately 13 km with slower ascent above. Enhanced $CO_2$ compared to the

stratospheric background can be detected up to 20 km. Mixing with older stratospheric air indicated by the decrease of measured $N_2O$ is found above ~17.5 km (400K potential temperature) (Vogel et al., 2023a).

Even when in our measurements no strong correlation between the relative abundance of the terrestrial refractory particle groups and the flight height was observed, a dependence on meteorology of the specific flight day was seen.

Bucci et al. (2020) showed that there was an enhanced convective influence in the second part of the StratoClim 2017 campaign

(flight 5 - 8). This becomes also visible in the increasing abundance of specific groups of refractive particles for flight 7 and 8.

For flight 7, the abundance of the other metals group is enhanced (on average from 3.3 % for flights 2- 6 to 13.3 %). At this day various small metallic/alloy particles (dominantly Cr, Mn and Zn-rich) from a specific ground region were transported into the ATAL. The ground regions from which the strongest input (or updraft) was observed indicate industrial high

temperature emissions in the Indo-Gangetic Plain/Northern India as source for these particles.



Flight 8 represents a special situation. In these samples the relative abundance of the Fe-rich and other metal group is increased in comparison to the results from all samples of the flights 2 – 6 (Table 2). Additionally, a large share of Hg-rich particles was detected in all samples of flight 8. These Hg-rich particles were almost absent in all other flight samples.

All three refractory particle groups are supposed to originate from fossil fuel burning or industrial high temperature processes.
The specific source(s) is (are) located in Northeastern India or Southern China. A detailed discussion of sources and pathways of the Hg-rich particles is given in the following chapter.

**4.4 Mercury rich particles**

Beside the so far discussed refractory particle groups, Hg-rich particles were detected in all samples of flight 8 (08/10/17). They were the most abundant particle group in flight 8 (1491 particles in total, on average 45 % of all refractory particles), while Hg-rich particles were almost absent in the eleven samples of flights 2 – 7 (in total only 8 Hg-rich particles).

Mercury is of high interest due to its toxicity and its ability to undergo long-range transport in the atmosphere. Natural sources include volcanic emissions, geothermal sources, and biomass burning. South Asia is known to show enhanced anthropogenic
Hg emissions (Kumari et al., 2015) from metal refining, incineration of waste, smelters, manufacturing units, as well as coal and oil combustion (Pirrone et al., 2010).

All Hg-rich particles during StratoClim2017 are very small with a $D_{P-max}$ of 25 nm (Figure 3). They are agglomerated to the surface of larger sulfate, nitrate, and/or organic particles. The small size is a clear indicator that these particles are formed by nucleation. The most probable terrestrial source for this of Hg-particle precursors will be the burning of fossil fuels (e.g., coal-
burning). Almost identical Hg-rich particles were found by Weinbruch et al. (2022) in samples directly taken at the stack of diesel and coal fired power plants on Svalbard. During fossil fuel burning mercury mainly passes into the gas phase as $Hg^0$. However, directly in the original plume parts of $Hg^0$ may adsorb as $Hg^{II}$ components on the surface of existing particles. For example, Seigneur et al. (1998) show that adsorption of $Hg^{II}$ species such as HgO and HgS on the surface of aerosol particles can account for up to 35% of total atmospheric mercury emissions.

In order to identify the specific mineralogical phase of these Hg-rich particles, additional measurements were performed in a

transmission electron microscopy (TEM). All 25 particles, examined by TEM-EDX ($D_P$ =15 – 35 nm) have an Hg:S atomic

ratio of about 1:1, while no other elements (beside carbon and oxygen) were detected. In high resolution, lattice planes could

be imaged within the Hg-rich particles (Fig.4a) and a diffraction image could be obtained in diffraction mode (Fig.4b). The

indexing confirmed HgS (cinnabar) as phase for all Hg-rich particles studied. Nucleation of HgS particles can only take place

under reducing conditions. Such conditions can exist in soils or during fossil fuel burning at ground but are very unlikely in

the UTLS.  Therefore, we conclude that the small HgS particles are already formed at the ground and then entered the UTLS

as primary particles with an appropriate updraft. Most probably the small HgS particles will agglomerate quite fast on the

surface of other existing particles (organics, sulfate, nitrates, or soot). This assumption is also supported by the fact that

cinnabar particles were only found in samples from one flight (flight 8). If these particles would have been formed in a

secondary process in the stratosphere or near the tropopause, they should rather be found as a general component in all flight

samples of StratoClim 2017 collected near the tropopause and especially in the samples of flight 3, which were taken in the

free stratosphere, what was not the case.

To identify a possible cinnabar source region at ground, back-trajectories were calculated based on the Chemical Lagrangian

Model of the Stratosphere (ClaMS) using high resolution ERA5 reanalysis (details see Vogel et al., 2023a,b). The trajectories

were calculated starting at the specific UTLS particle sampling time/location back to the start of the monsoon season

(06/01/2017). Data endpoints are shown for all trajectories reaching the model boundary layer by then. Further on, in order to

identify the position of strongest uplift of air along the back-trajectories, the mean location of the strongest change of potential

temperature along the back-trajectories (running mean over 6 hours) was calculated. The results for flight sample 8.5

(exemplary for all very similar graphs of the flight 8 samples) is shown in Figure 5. Frequency distribution (fd) of air mass

origins show that the possible source regions for an entry of terrestrial particles is located in the Indo-Gangetic Plain,

respectively Northeastern Indian Subcontinent and in Southern China. Anthropogenic emissions in the Indo-Gangetic Plain

are higher compared to other regions in India caused by the dense concentration of industries as well as by the very high

population density in this area. Air masses transported from the Indo-Gangetic Plain (or passing it) uptake the anthropogenic

emissions and were mainly uplifted along the southern edge of the Himalayas or by strong convection to UTLS altitudes.

These specific source regions as well as favorable meteorological conditions for particle transport from ground to UTLS is also supported by Bucci et al (2020), who studied the impact of deep-convective transport on ATAL during the StratoClim 2017 campaign.

Bucci et al. also stated that flight 8 captured some very intense overshoots and convective outflows from exceptionally fast (less than an hour) and localized plumes.


India and China are among the largest emitters of atmospheric mercury in the world (Jetashre, 2022; Wu et al., 2006). For example, the Indian Jharia region, which is directly within the identified source-region, with the Jharia coal fields (23.75°N 86.42°E) produces most of India's coal. Jharia coal mines are India's most important storehouse of prime coke coal and consists of 23 large underground and nine large open cast mines. Furthermore, there are persistent smoldering coal field fires in this

region for more than a century. The Hg concentration of the coal is higher than world average and the coal field fires are known to be a source for Hg pollution in the mining area (Raj et al., 2017). According to Nadudvari et al. (2022), HgS particles are formed above underground coal deposit fires and thermally affected waste dumps from hard coal mining due to the reducing environment in the bituminous surface layer. We therefore conclude that the detected cinnabar particles in flight 8 originate from industrial coal burning and underground coal fires in Northern India or Southern China. Cinnabar may have formed either

under or at ground under reducing conditions and instantly adsorbed on the surface of other aerosol particles transported into the UTLS by a strong updraft, maybe during the intense overshoot events observed during flight 8.

In all previous literature a different origin of particular $Hg^{II}$ components in UTLS is inferred. Murphy et al. (1998) discussed Hg-rich particles from UTLS and stratospheric aircraft experiments. Hg was detected in a large number of MS particle spectra during one flight leg near the tropopause south of Houston to 10°N. In contrast, no mercury was found during flights in a

remote continental surface site (Idaho Hill, Colorado) and a remote marine surface site (Cape Grim, Tasmania). During different aircraft campaigns in the tropics and middle latitudes, Murphy et al. (2006) detected Hg-containing particles close to the tropopause, while no Hg-containing particles were detected below 5 km height. They concluded that particular $Hg^{II}$ most likely originates from oxidization of gaseous $Hg^0$ in the lower stratosphere and not from a primary terrestrial Hg-particle source. This conclusion is also supported by Lyman and Jaffe (2011). During aircraft measurements at 6 - 7 km altitude, $Hg^{II}$ was

positively correlated with stratospheric tracers (ozone and potential vorticity), indicating that Hg[II] increased with increasing stratospheric influence.

The Hg chemistry in the atmosphere is quite complex and subject of current scientific research. Excellent overviews of possible mercury oxidation pathways in the atmosphere are e.g., Schroeder and Munthe (1998), Holmes et al. (2010), Shah et al. (2016), Obrist et al. (2018), and Lyman et al. (2020). They summarize the possible role of OH, $O_3$, bromine, photochemistry and

aqueous-phase reduction. Gratz et al. (2015) reported results from the 2013 Nitrogen, Oxidants, Mercury and Aerosol Distributions, Sources and Sinks campaign, which supported the role of bromine as the dominant oxidant of mercury in the upper troposphere and the importance of subtropical anticyclones for the formation of Hg[II].

This is worth highlighting, because during the StratoClim flight campaign in 2017 Adcock et al. (2021) detected enhanced bromine values in the UTLS.

Up to now it has been concluded that the source of Hg[II] particles in the UTLS can be completely attributed to stratospheric oxidation of gaseous Hg[0]. We observed that also primary Hg[II] particles are directly transported into the ATAL. This direct transport may be favored by quick convective outflows of very fast and localized plumes as observed during StratoClim flight 8. This implies that for modelling of the transport of Hg components into the ATAL, primary Hg[II] emissions on ground must also be considered.

This transport from the ground could also be responsible for the Hg-particles described in Murphy et al. (2006) for a lower-stratospheric sample. Even when no mineralogical phase information is given by Murphy et al. (2006) their observed Hg-rich particles seem to be identical with the HgS (cinnabar) particles observed by us. Size (10-20nm in diameter), mixing-state (attached to sulfate particles), Hg:S ratio, and the beam sensitivity in STEM (volatilization under STEM conditions in a few seconds) were identical to our cinnabar particles found during StratoClim. Thus, the presence of HgS particles in UTLS seems

not to be limited to the specific conditions within the ATAL during the 2017 monsoon anticyclone.

## 5 Conclusion

It was shown that within the 2017 Monsoon anticyclone there is a predominantly terrestrial input of refractory particles into the ATAL.

In contrast to prior modeling studies, we found that refractory particles (including desert dust) play only a minor role in the total composition of aerosol particles within the ATAL. In SEM measurements about 2% by number of the typical ATAL particles (main components: ammonium, sulfate, nitrate, and organics) show in SEM visible inclusions/agglomerates of refractory particles. The main components within the refractory particles were silicates and NVOC. In addition, Fe-rich particles, other metal-rich particles (Mn, Cr, Zn) and extraterrestrial particles were found, as well as some small amounts of soot, Ca-, Cl-, and Al-rich particles.

In general, most refractory particles found are very small. The maximum of the $dN/d\log D_P$ distribution is at ~125 nm. This also means that, beside some very small terrestrial soil particles, nucleation processes are the predominant source. For most refractory particles these are mainly anthropogenic combustion processes (coal burning, biomass burning, industrial processes). For the NVOC additionally secondary atmospheric processes are important.

The variability of the relative number abundance of individual ground emitted refractory particle groups was quite low during StratoClim 2017 for most sampling days and at different flight altitudes between ~ 12 - 19 km. This suggests that there was generally a roughly uniform background composition for the terrestrial refractory particles in the ATAL and also just beyond the tropopause (lower stratosphere). Extraterrestrial refractory particles play a larger role above the tropopause, within the ATAL their relative abundance is low.

During flight 7 and 8, additional refractory particles were detected. These particles originated from an additional input from special anthropogenic ground sources and were rapidly transported into the UTLS under enhanced convective influence. The ground regions from which the strongest input (or updraft) was observed indicate industrial emissions in the Indo-Gangetic Plain for flight 7, which additionally introduced various metals/metal oxides into the ATAL here. For flight 8, detected cinnabar (HgS) particles indicate an additional input from coal combustion, most likely from Northeastern India or Southern China. The effective transport of these particles into the ATAL may be due to some very intense overshoots and convective outflows from very fast and localized plumes during this flight.



The direct input of primary $Hg^{II}$ particles into the tropopause region represents a previously undescribed distribution pathway for atmospheric Hg, since up to now only the ground input of gaseous $Hg^0$ and oxidation to $Hg^{II}$ in the stratosphere was proposed.

**Data Availability**

The complete data set is available for the community and can be accessed by request to Martin Ebert (mebert@geo.tu-darmstadt.de) of the Technical University Darmstadt.

**Author contribution**

R.Weigel- potential temperature, CPT data analysis and classifiction of particle origin, data discussion and interpretation

S.Weinbruch- manuscript data interpretation and discussion

L.Schneider- SEM data evaluation, interpretation and discussion

K.Kandler- data evaluation/classification, interpretation and discussion,

S.Lauterbach- TEM measurements and TEM data interpretation/discussion

F.Köllner- Classification, comparison and interpretation of refractory (SEM and MS) particle data

F.Plöger- field experiment data, data discussionand interpretation (meteorology/ATAL/UTLS)

G.Günther- field experiment data, data discussion and interpretation (meteorology/ATAL/UTLS)

B.Vogel- back trajectory analysis and air mass origin discussion

S.Borrmann- manuscript data interpretation and discussion

**Competing interests**

The authors declare that they have no conflict of interest.

## Acknowledgements

This work was supported by TPChange (The Tropopause Region in a Changing Atmosphere)-DFG TRR301. The Nepal aircraft campaign was conducted within the project STRATOCLIM sponsored by the European Union Seventh Framework Programme (FP7/2007-2013, grant no. 603557). The StratoClim project was financially also supported by the German "Bundesministerium für Bildung und Forschung" (BMBF) under the joint ROMIC-project SPITFIRE (grant no. 01LG1205A) as well as by the European Union Seventh Framework Programme (FP7/2007-2013, ERC grant no. 321040-Excatro). The presented work includes contributions of the NSFC–DFG 2020 project ATAL-track (BO 1829/12-1 and VO 1276/6-1). The authors thank the M-55 Geophysica team and the MDB (Myasishev Design Bureau, Moscow, Russia) for planning and carrying out the flights.

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



**Table 1: Parameters for particle samples (coarse: equivalent projected area diameter > 0.4 μm; fine: equivalent projected area diameter 0.04 – 0.4 μm).**

| flight nr. | date mm/dd/yy | sample nr. | particle size fraction | sampling start* (UTC) | sampling duration [min] | flight height [km] |
|---|---|---|---|---|---|---|
| 2 | 07/29/17 | 2.1 | coarse | 3:22 | 13 | 12.5 – 15.0 |
| 3 | 07/31/17 | 3.6 | coarse | 5:09 | 13 | 19.8 |
| 4 | 08/02/17 | 4.5 | coarse | 10:15 | 15 | 17.5 – 18.0 |
| 5 | 08/04/17 | 5.2 | coarse | 4:35 | 18 | 16.3 – 17.0 |
| | | 5.2f | fine | 4:35 | 18 | 16.3 – 17.0 |
| 6 | 08/06/17 | 6.5 | coarse | 10:05 | 15 | 16.2 |
| 7 | 08/08/17 | 7.1 | coarse | 4:29 | 17 | 12.6 – 14.3 |
| | | 7.1f | fine | 4:29 | 17 | 12.6 – 14.3 |
| | | 7.4 | coarse | 5:29 | 17 | 13.0 – 18.0 |
| | | 7.4f | fine | 5:29 | 17 | 13.0 – 18.0 |
| 8 | 08/10/17 | 8.1 | coarse | 9:19 | 17 | 12.0 – 16.0 |
| | | 8.1f | fine | 9:19 | 17 | 12.0 – 16.0 |
| | | 8.2 | coarse | 9:40 | 17 | 16.0 – 17.0 |
| | | 8.2f | fine | 9:40 | 17 | 16.0 – 17.0 |
| | | 8.4 | coarse | 10:20 | 17 | 17.0 |
| | | 8.5f | fine | 1040 | 17 | 17.0 |
| | | 8.6 | coarse | 11:01 | 17 | 17.1 – 17.8 |

* local time = UTC + 5.45 h





**Table 2: Number of analyzed refractory particles/inclusions after evaporation of volatile matrix.**

| sample number | 2.1 | 3.6 | 4.5 | 5.2 | 5.2f* | 6.5 | 7.1 | 7.1f* | 7.4 | 7.4f* | 8.1 | 8.1f* | 8.2 | 8.2f* | 8.4 | 8.5f* | 8.6 | total |
|---|---|---|---|---|---|---|---|---|---|---|---|---|---|---|---|---|---|---|
| extra-terrestrial | 1 | 16 | 35 | 25 | 10 | 33 | 3 | 19 | 52 | 28 | 11 | 8 | 12 | 11 | 15 | 3 | 14 | **296** |
| silicate | 15 | 62 | 71 | 76 | 22 | 29 | 42 | 306 | 87 | 88 | 35 | 241 | 15 | 379 | 43 | 54 | 57 | **1622** |
| Ca-rich | 2 | 3 | 0 | 2 | 6 | 3 | 2 | 18 | 3 | 3 | 7 | 2 | 5 | 30 | 1 | 0 | 7 | **94** |
| Cl-rich | 1 | 1 | 2 | 3 | 2 | 0 | 0 | 0 | 8 | 3 | 1 | 1 | 4 | 7 | 6 | 2 | 6 | **47** |
| Fe-rich | 1 | 4 | 3 | 9 | 7 | 6 | 10 | 61 | 11 | 14 | 14 | 28 | 35 | 69 | 24 | 3 | 16 | **315** |
| Al-rich | 1 | 3 | 0 | 0 | 0 | 0 | 2 | 8 | 2 | 2 | 2 | 1 | 3 | 8 | 3 | 3 | 1 | **39** |
| other metals | 0 | 2 | 4 | 9 | 5 | 3 | 39 | 55 | 12 | 13 | 25 | 41 | 23 | 45 | 10 | 6 | 10 | **302** |
| soot | 0 | 0 | 0 | 1 | 13 | 0 | 1 | 20 | 1 | 18 | 5 | 6 | 1 | 19 | 8 | 23 | 8 | **124** |
| C-rich | 5 | 39 | 21 | 26 | 19 | 8 | 4 | 61 | 14 | 24 | 53 | 66 | 12 | 48 | 35 | 11 | 85 | **531** |
| Hg-rich | 0 | 0 | 2 | 0 | 3 | 0 | 0 | 1 | 2 | 0 | 303 | 26 | 38 | 90 | 454 | 288 | 292 | **1499** |
| others | 2 | 4 | 4 | 5 | 2 | 1 | 22 | 32 | 5 | 14 | 11 | 11 | 6 | 35 | 3 | 2 | 5 | **164** |
| total | **28** | **134** | **140** | **156** | **89** | **83** | **125** | **581** | **197** | **207** | **467** | **431** | **154** | **741** | **602** | **395** | **501** | **5033** |

*f = fine stage sample 0.04 – 0.4 µm equivalent projected diameter





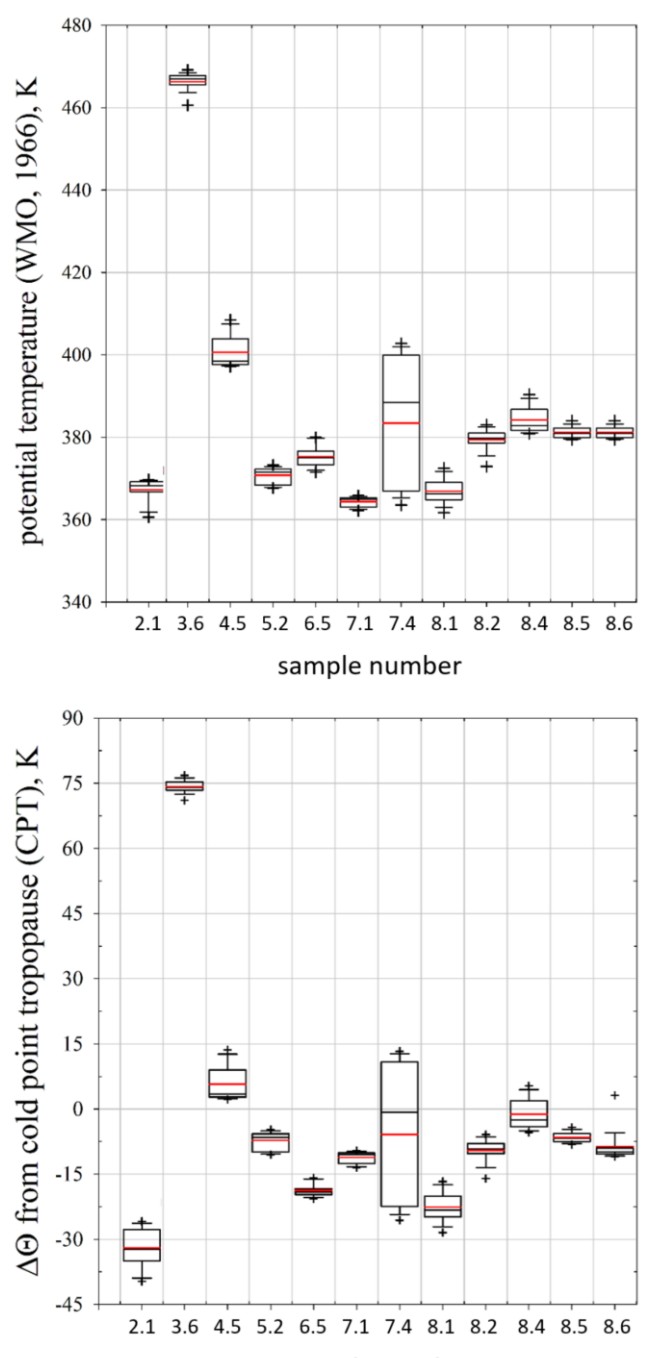

**Figure 1: Boxplot of the absolute potential temperature Θ (upper), and of potential temperature Θ difference to the 1 Hz calculated Θ-level of the cold point tropopause (CPT) throughout sampling period (lower).**



**Figure 2: Relative number abundance of refractory aerosol particles/inclusions within the 17 UTLS flight samples from 7 different flights during StratoClim 2017 (f = fine stage).**





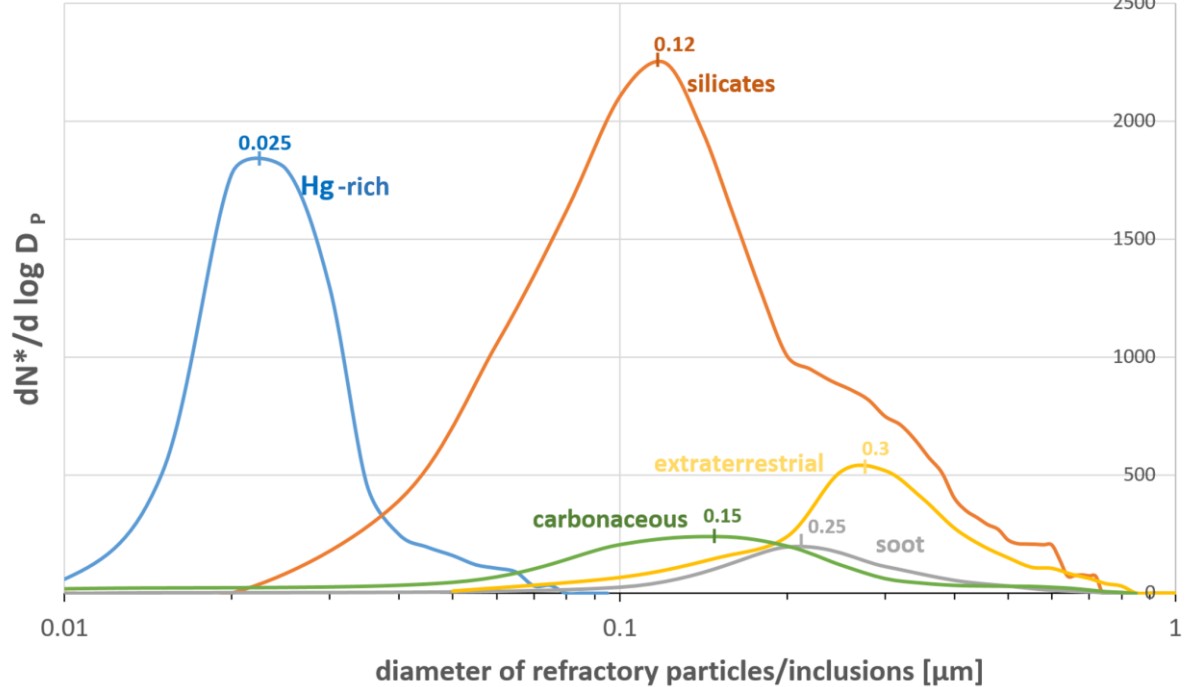



**Figure 3: Smoothed relative size distribution of 5 refractory particle/inclusion groups (dN\* = total number of analyzed refractory particles/inclusions of specific group within all 17 analyzed samples.).**





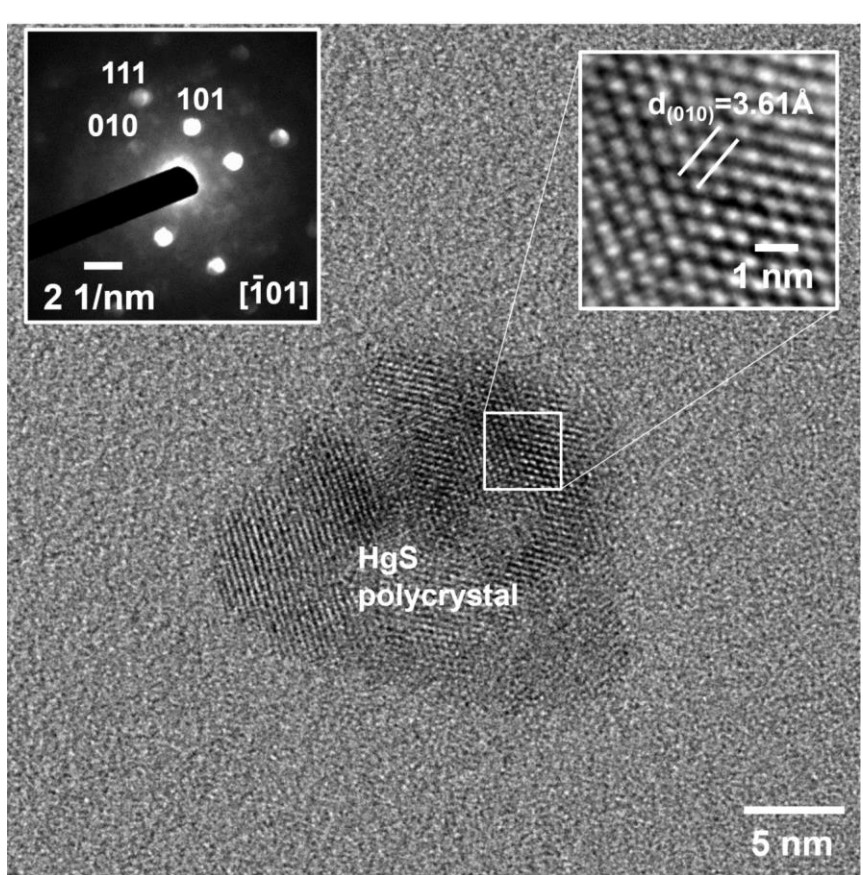

**Figure 4: High-resolution TEM image of a 20 nm small poly-cristalline HgS (cinnabar) particle. Inset left) corresponding convergent beam electron diffraction (CBED) pattern. Inset right) Invers Fast Fourier Transformed (IFFT) image from the area indicated by the square.**





Figure 5: a) Frequency distribution (fd) of air mass origins at the model boundary layer for sample 8.5. Back-trajectories were calculated using ERA5 reanalysis back to the start time of the monsoon season (06/01/2017). Only back-trajectories are considered reaching the model boundary layer by then. b) Frequency distribution (fd) of the mean location of the strongest change of potential temperature along the back-trajectories (running mean over 6 hours) indicating the position of strongest uplift of air along the trajectory.