# Peer review of "Characterization of refractory aerosol particles collected in the tropical UTLS within the Asian Tropopause Aerosol Layer (ATAL)"

_EGUsphere, 2023_

## Author Response (AR1)

**Answers to Reviewer 1**

1. *General: The manuscript very much needs a figure showing vertical profiles of either the absolute or relative concentrations of various aerosol types. Either potential temperature, distance from the tropopause, or ozone could be used as a vertical coordinate. This could be put in somewhere around line 400. Figure 1b could be deleted or moved to supplemental material to make room; it largely duplicates Figure 1a.*

**The offline analysis of aerosol particle flight samples (respectively the airborne particle sampling) provides only a poor time resolution (and thus often also height resolution). In most flights, the height was considerably varying during particle sampling. Thus, most samples cannot be assigned to one specific height but originate from a height range, which is given in Table 1. Beside the abundance of extraterrestrial material there is no clear observable trend in the relative abundance of refracory particle groups in dependence from flight height. Due to both points we refrained from showing vertical profiles of concentrations of various aerosol types.**

2. *Lines 25-30: These conclusions are not supported; see below.*

**We have deleted these conclusions in the manuscript (see point 8 below)**

3. *Line 71-73: This paragraph keeps saying it is about the UTLS but in fact it is about the lower stratosphere. In particular, transport of refractory particles is much more important in the upper troposphere. This paragraph would be OK if "UTLS" is just changed to "lower stratosphere".*

**Changed as requested (line 59)**

4. *Section 2: What I am not clear on after reading this is what will be measured in the situation that there is a refractory species dispersed in a particle rather than being originally present as a distinct inclusion. Will it be missed? One example is that I kept wondering about: where is all the potassium? A significant minority of particles in the ATAL will be from biomass burning. Those particles contain a percent or a few percent potassium. 1% of the volume of a 200 nm particle is 43 nm, much more than the 25 nm Hg particles. Would this be missed?*

**The decisive factor for this question is whether the element is homogeneously or heterogeneously distributed within the particle. The EDX system has an excellent absolute detection limit but a relatively weak relative detection limit. For small particles (~smaller 200nm) the spectrum background and in this way the detection limit is even worse, so that often only major elements can be detected. Therefore, an external 25 nm Hg-rich particle yields a clear Hg-peak, but an EDX spectrum of a 200nm biomass burning particle often show unambigious C and O peaks only.**

**My time-intensive SEM measuring methodology to detect all refractory particles has proven to work very well for all particle types with one limitation. The amount of small carbon-rich refractory particles (organics, biomass burning and soot) will be underestimated. We have clarified this point in the manuscript. (see also answer to question 6 of reviewer 2)**

5. *Lines 253-255: The extraterrestrial particles are an odd size: too big to be the residuals from evaporated sulfuric acid particles with meteoric material, too small to be cosmic spherules. That doesn't mean the data are wrong but it needs some justification.*

**The unambiguous detection of extraterrestrial material in the SEM is difficult. We cite the related classification and emphasise the uncertainties.**

**Line 317-329:**

**"For the classification of stratospheric particles often a broader classification is applied as it was used here. For example, in the NASA cosmic dust catalogue (Warren et al., 2011) not only chondritic compositions are classified as "cosmic", but also compositions strongly modified by ablative heating or melting during passage through the atmosphere. Following the NASA definition (which is not complete applicable for this work as it is defined for optical microscopic data of large super-µm particles) all particles from our extraterrestrial-, silicate-, and Fe-rich groups would be classified as "cosmic". The NASA definition is appropriate for stratospheric particles, collected well above the tropopause, where little terrestrial admixture is present. For particles collected below the tropopause and specially inside the ATAL, where increasing entry of terrestrial silicate material occurs, an unambiguous attribution of Mg, Si and/or Fe-containing particles to a terrestrial or extraterrestrial source is difficult. Thus, the classification of extraterrestrial particles in the upper troposphere used here is associated with a substantial uncertainty. Furthermore, it has to be considered that after ablation processes a part of the incoming extraterrestrial material (iron) may be "dissolved" in sulfuric acid droplets (discussion in**

**chapter 4.3). If the total extraterrestrial input should be estimated, both, the dissolved fraction and the**

**refractory particles/inclusions must be considered. "**

**In summary, we have classified extraterrestrial particles as described above and simply present the**

**results.   We are not aware of published data on the size distribution of extraterrestrial particles, which**

**are comparable with our measurements.**

6. *Lines 282-290. Where is all of the small soot? The number size distribution of black*
   *carbon in the upper troposphere peaks at about 100 nm (Schwarz et al., 2006) with very*
   *few as large as the 250 nm mean diameter quoted here.*

As clarified in the manuscript we underestimate small carbonaceous particles (C-rich particles
– see answer 6 to reviewer 2- and soot).

We agree…Clarified text, line 299-305:

**"Soot can be recognized in SEM by its typical morphology of agglomerates of spherical
primary particles. As the lateral resolution of the SEM is limited and the characteristic
morphology described is often no longer observable for very small soot particles (DP ≤ 100
nm), this morphological criterion cannot be applied for these particles. Thus, the abundance
of the soot group represents a minimum share and the received average diameter of this
group ($\overline{D}_P$ = 250 nm) will be too high."**

7. *Line 331. I don't see any justification for assuming that Al-rich particles come from rocket*
   *exhaust rather than Al-rich minerals.*

We agree…Clarified text (line 352):

**"Al-rich particles are supposed to be mainly aluminum oxide and originate either from Al-rich
minerals or from solid rocket fuel exhausts (Mackinnon et al., 1982; Cziczo et al., 2002)."**

*8.(Main criticism) "I am not convinced by one of the main conclusions of the manuscript, that
coal burning is a source for nanoparticles of HgS in the Asian tropopause aerosol layer (ATAL)."*

*Summarized criticism: HgS from coal burning is not matching to these findings:*

  *- no increased soot content in flight 8 samples*

*- no other Hg species like HgCl$_2$ in flight 8 samples*

*- no high CO values in flight 8 samples*

**We still believe that the formation of HgS in the UTLS or lower stratosphere is unlikely. Furthermore, it can be excluded that a chemical transformation of a previously different Hg phase in the electron microscopes to sulphide occured. We checked this carefully in our experiments. In addition, formation of HgS induced by electron beam interactions is not described in literature.**

**We agree with your criticism that a strong convective input of coal combustion during flight 8 should lead to increased levels of soot, CO, and other Hg phases (e.g., HgCl$_2$) but is not observed.**

**We have therefore removed the statement that the HgS particles are clearly from coal burning and have rewritten chapter 4.4 (starting line 458). We now discuss additional sources of these particles.**

**We have added the three references proposed by reviewer 1:**

Keun-Ok Lee et al., ACP, 2021, 10.5194/acp-21-3255-2021

Peng et al., Mercury speciation and size-specific distribution in filterable and condensable particulate matter from coal combustion, Science Total Environ., 2021.

Srivastava et al., Control of mercury emissions from coal-fired electric utility boilers, ES&T, 2006

**Answers** to the "*questions of reviewer 2*":

    1.) „*I suggest adding 3D flight pattern plots and back trajectory plots for each flight. It is hard for me to understand your flights and the potential source of your samples.*"

As there have been some requests for more extensive data about the flights, we added to the already shown table 1 (flight height) and figure 1a+b (potential temperature) some additional graphs/tables in the supplement. In Figure S1 we present all flight patterns of all particle collections. Additionally we have added Figure S2a+b+c, in which we show the results of the back-trajectory analysis for all particle samples. Up to now, we had shown this only exemplary in the main manuscript for sample 8.5.

    2.) "*Moreover, I don't understand how you label your samples since they are not integers.*"

To make the sample labels more clear, we have added an explanation:

Line 155:" …The individual sample labels follow this sequence. For example, sample 8.5 corresponds to the fifth particle sampling of flight 8."

    3.) "*There is a lot of information missing for your home-built instruments. For example, what is the size range of COPAS? Did you have a dryer and impactor or PM inlet in front of the aerosol inlet to remove particles larger than the upper limit?* "

To give more information about the COPAS, we have added this section, line 118-131:

"Particle samples were taken by the inlet line of COPAS (Condensation Particle counting System; Curtius et al., 2005; Weigel et al., 2009; Borrmann et al., 2010) with a Y-shape manifold. According to Weigel et al. (2009) the inlet efficiency is comparable with the inlet system characterized by Hermann et al. (2001). For the super-isokinetically operated, predominantly isoaxially aligned aerosol inlet, it is determined that the aspiration, transmission and transport of particles with diameters (Dp) of up to one µm through the aerosol lines to the instruments occurs without significant losses. For submicron particles, the transmission efficiency of the COPAS inlet is ≥ 90 %. The inlet performance rapidly deteriorates for increasing particle diameters and is between 30 – 40 % for particles with 4 µm diameters, and ≤ 5 % for particles larger than 6 µm. Even larger particles (Dp > 10 µm) are in general unlikely to be aspirated by the inlet despite superisokinetic operation. The sample flow is branched towards the multi cascade impaction system Multi-MINI (Ebert et al., 2016) after about three quarters of the entire aerosol line between the inlet system and the COPAS entry (after 45 cm of a total of 55 cm long, quarter-inch stainless steel aerosol line with connections of electrically conductive tubing). Downstream of this flow splitter, the sample stream is passed via an approximately 20 cm long stainless-steel tube (quarter-inch diameter) towards

the MULTI-MINI impactor. The exhaust air from the MULTI-MINI impactor pump is led into the exhaust line shared with the COPAS system and released outside the aircraft."

4.) *"Did you do any lab evaluation of the performance of the MULTI-MINI impactor system?"*

Twelve classic cascade jet impactors are installed in the Multi-Mini. Details on the design and performance of these impactors can be found in Kandler et al., 2007 and the references therein. The performance of the rinsing unit of the Multi-Mini was successfully tested in blank trials (also during this campaign/ described in the manuscript) and during a prior campaign (Ebert et al., 2016).

Ebert, M., Weigel, R., Kandler, K., Günther, G., Molleker, S., Grooß, J.-U., Vogel, B., Weinbruch, S., and Borrmann,S.: Chemical analysis of refractory stratospheric aerosol particles collected within the arctic vortex and inside polar stratospheric clouds, Atmos. Chem. Phys., 16, 8405–8421, https://doi.org/10.5194/acp-16-8405-2016, 2016.

Kandler, K., Benker, N., Bundke, U., Cuevas, E., Ebert, M., Knippertz, P. , Rodriguez, S., Schütz, L., Weinbruch, S.: Chemical composition and complex refractive index of Saharan Mineral Dust at Izana Tenerife (Spain) derived by electron microscopy. Atmos. Environ. 41, 8058-8074, 2007.

5.) *"Are there any references for ERICA-LAMS? Please provide details about how it works. Without knowing that, it is hard to understand the difference between the two technologies. Also, is this bulk aerosol measurement or individual aerosol measurement technique?"*

We have added some more information to the *individual aerosol measurement technique ERICA-LAMS, line 224-240*:

line226..: "During StratoClim 2017, simultaneous aerosol mass spectrometric measurements with the ERICA-LAMS (Laser Ablation Mass Spectrometer) instrument were conducted. ERICA-LAMS can measure refractory and non-refractory aerosol components by laser ablation and ionization technique followed by time-of-flight aerosol mass spectrometry. The instrument has been described before in detail (Hünig et al., 2022; Dragoneas et al., 2022) and is only briefly

reviewed here. Particles enter the system through a pressure-controlled inlet (Molleker et al., 2020). The following aerodynamic lens focuses particles into a narrow beam. The particles are optically detected by scattering light when passing through two laser beams. This setup provides the particle's time of flight and its velocity. By using a calibration with particles of known size, density, and shape, we can derive the aerodynamic diameter. The $d_{50}$ cut-off of the ERICA-LAMS is thus limited by the optical particle detection efficiency to 180 nm. The detected and sized particles are ablated and ionized by a single triggered laser shot (wavelength = 266 nm) and the ions are guided into a time-of-flight mass spectrometer. As result, the ERICA-LAMS provides bipolar mass spectra and size of individual particles.

In these measurements it was determined that between 20 and 50 % of all measured particles (by number) include refractory material, depending on altitude (Appel et al., 2022). The values between the two techniques differ, obviously, for different reasons. The refractory number abundance of 2% determined by SEM/EDX measurements refers to refractory heterogeneous inclusions within ATAL particles only (see chapter 2.4) while the MS derived value of 20 – 50 % reflects the proportion of particles that provide any refractory signal including dissolved refractory elements, which may play also an important role (e.g., for extraterrestrial material, see Schneider et al., 2021 and discussion in chapter 4.3)."

6.) *Are your results based on the EDX spectra of some portion of the particle or EDX mapping? If yes, how much % of the area of the particle was covered? Do you think that is representative since you might only get partial chemical information about the particle? Also, how do you determine particle size? Is this area equivalent diameter? What are the parameters you used for each class? Is that based on the weight or elemental %? How did you develop your classification method? From literature or K-mean clustering? I suggest adding a table to show how you classify each particle. Why don't you consider refractory organic aerosols (e.g., ELVOC) as refractory particles? These can be very important compositions in the tropopause aerosol population.*

The classification of the 5033 individual particles measured (Table 2) is based on the main elements of these particles, derived from individual EDX measurements. Based on the small size of the particles (50% are smaller than 100 nm), in almost all cases the entire refractory particle is excited during an individual SEM/EDX measurement (the penetration depth/ excitation bulb at 15 kV is in the μm-range) and the main elements of the entire particle are received.

Therefore, the received main elements contained for the small refractive particles are representative for the whole particle. Furthermore, no statements can be made from these SEM/EDX measurements about minor elements, the internal mixing state of these small

refractory particles, or about their mineralogical phase composition. This was only possible applying additional TEM measurements for some individual particles.

All given particle diameters are equivalent projected area diameters.

Due to the fact that often only the main elements can be detected in the EDX spectrum of the small particles, the classification can usually only be based on these main elements (Si-rich (silicates), Al-rich, Ca-rich, Cl-rich, Fe-rich, Hg-rich, Si+Mg or Fe+Mg (extraterrestrial), sum of other metals (other metals)).

As these classification rules are quite simple we prefered presenting them in the text (Chapter 3.1) and not in a separate table.

In fact, the situation is somewhat more complicated for the C-rich particles.

In almost all individual EDX particle measurements the foil of the TEM mesh is also excited, so that a small amount of C is always visible in the spectrum (the C signal becomes stronger the smaller the particle is). This is also the reason why the determination of the proportions of C-rich particles such as soot and refractory organic aerosols is associated with larger uncertainty. While soot particles can often be clearly identified due to their characteristic morphology, it can be assumed that the proportion of refractory organic particles present is underestimated. Volatile and refractory organic particles are of immense importance within ATAL, but I can only make limited statements about this component.

To make this point clearer, we have added, line 306-310:

" As the TEM mesh is also excited during the analysis of very small particles, the C background in the spectrum is significantly increased in many of these spectra, which generally makes the clear identification of small carbonaceous particles (C-rich and soot) more difficult.  Therefore, the proportion of carbonaceous particles shown in this work is only a minimum proportion. Other studies have clearly demonstrated the special importance of organic particles in ATAL (e.g., Appel et al., 2022)."

7.) *Most of your classes sound like different types of dust (extraterrestrial material, silicates, Ca-rich, Fe-rich, Al-rich). It is very difficult to validate your classification without SEM imaging and EDX spectra. Also, if your back trajectory plots show a significant contribution from the boundary layer, then I think your refractory particles are mostly dust.*

The categorisation into our 11 classes is primarily based on the chemical composition of the particles and does not include a source classification (exception: extraterrestrial and soot).

There are basically 3 main possible sources for the refractory UTLS particles: extraterrestrial material, natural soil material (short: dust) or particles from nucleation/combustion

processes. While soil material certainly makes up a large proportion (e.g. silicates/Ca-rich), we assume that nucleation/combustion processes (e.g. Fe-rich, other metals) are also a source of very small refractory particles. The assignment of the detected particle groups to these specific source-types is uncertain as discussed in detail in the manuscript.

The SE images of 100nm or smaller particles show in almost all cases no characteristic details (because of limited resolution) as it is the case for larger particles. In this way I am afraid there is no gain in knowledge by showing them.

Minor comments

1. L106-108, "The absolute …. (upper part)." I would suggest having a SI plot to show temperature and pressure, and another SI plot to show time series of temperature and pressure for each sample.

**We prefer not to show SI-units, but the absolute potential temperature Θ and potential temperature Θ difference to the 1 Hz calculated Θ-level of the cold point tropopause (CPT) as most cooperating groups requesteted this kind of presentation.**

2. L134-135, "In this study … 400 nm)." 400 nm and 40 nm are the 50% cut-off size, not the boundary of each stage.

**We use the term "cut-off"**

3. L136-137, "A purge flow … "  Is this at the ground level or UTLS? Did you add a dryer and a filter in front of the purge flow? Ambient air might introduce additional contamination.

**In our impactor, each sampling is started after a few minutes of "purging" the sampling line with new air. This procedure has proven to avoid carry over effects from old air-masses. The "purging-air" is taken from the surrounding air (so identical with sampled air).**

4. L161-162, "A major challenge … boundary layer." In future studies, you can put a filter in front of the sampling inlet to verify and quantify the contaminations in the sampling line.

**We have done so successfully in some blind measurements in test flights, but unfortunately we had no technical solution to do this automatically during the campaign in the Geophysica, where no scientific operator is onboard. We hope that we will have this feature for future campaigns.**

5. L175-181, "First, all … excluded." This part is unclear to me. I do not understand how you identified contamination since you did not provide any representative SEM image and EDX

spectrum. These steel-like and large particles still might be real particles. Also, when you say exclude samples with too few particles, how many are you considering too few?

**In a jet impactor all particles are concentrated in a small impaction spot on the substrate. Only samples with a visible impaction spot were selected for analysis.**

**Corrected text, line 182-187:**

**"To minimize the risk of interpreting artefact particles as real refractory components, all samples were sorted out for which errors were recorded during the sample change or afterwards during handling that could have led to possible contamination (in total 5 samples).**

**Further on, samples were excluded when too few particles were found on the substrate as in this case it cannot be guaranteed that the number of potential refractory artefact particles is negligible. Based on this criterion, many of the received particle samples from flights #2, #3, #4, #5, and #6 had to be excluded.**

**Finally, samples which were selected for analysis but with less than 25 refractory particles found were also excluded for final data analysis due to statistical reasons.**

6. L200-201, "Therefore, … bombardment." How did you get the number of volatile particles? Based on your COPAS data?

**The 270.000 volatile particles were vaporized (and counted) in the SEM. This is now clarified in the text (line 206-212).**

7. Hünig et al., 2022 and Dragoneas et al., 2022 are not in the references list.

**We added the references, thank you.**

8. Section 4.2. This section is unclear to me what you are trying to discuss. I do not think you can get the absolute concentration of refractory particles unless you did comprehensive calibration of your sampling system, which includes collection efficiency of the impactor, particle loss in the line and impactor, density of each particle type, etc.

**We agree that the direct determination of the absolute concentration of the refractory particles would be very inaccurate. Therefore, we only used our ratio of refractive particles/volatile particles. The absolute refractory particle concentrations were estimated using this ratio and the total aerosol concentration determined by Appel et al. (2022).**

**This in now clarified in the text, line 378-391:**

9. L347-379, "Only a small … above the tropopause." I am not convinced in this part. First, it is not clear how you define ground-emitted and extraterrestrial particles based on my previous comments. Your discussion about the source is also not convincing to me.

**We tried to clarify our statements (see points above). Again, particle classification is based on major element composition and in the case of soot on morphology. Assignment of the resulting particle groups to sources is uncertain as discussed in detail in the manuscript. We tried to clearly disclose these uncertainties and discuss our interpretation in the context of existing literature.**

10. L394-395, "The presence of … silicates." This part is not clear to me. You should also be able to collect droplets and see Fe and Mg in the droplet's residual. If you use a dryer, you should collect dry Fe and Mg particles after removing moisture.

**After evaporating the volatile (sulfate/nitrate/organic) matrix in the SEM, no Fe and Mg-rich particles were found in the visible residuals. The reason for this stays unclear.**

---

## Author Response (AR2)

There were two open points:

1. **Identified particle classes need to be illustrated with representative SEM images, EDX spectra, and average elemental percentage, which could be either included in SI file or incorporated into the manuscript.**

SEM images, EDX spectra and elemental percentage of the classified particle groups are added in Figure S4 in the supplement. Additionally we have added Figure S3, which shows the complete classification criteria.

2. **Figure 3 requires better artwork.**

We have improved Figure 3.